# Graphical Representation of Cavity Length Variations, Δ*L*, on s-Plane for Low-Finesse Fabry–Pérot Interferometer

**DOI:** 10.3390/s25072182

**Published:** 2025-03-29

**Authors:** Alex Guillen Bonilla, José Trinidad Guillen Bonilla, María Eugenia Sánchez Morales, Héctor Guillen Bonilla, Maricela Jiménez Rodríguez, Antonio Casillas Zamora

**Affiliations:** 1Departamento de Ciencias Computacionales e Ingenierías, CUVALLES, Universidad de Guadalajara, Carretera Guadalajara-Ameca Km. 45.5, Ameca 46600, Jalisco, Mexico; alex.guillen@academicos.udg.mx; 2Departamento de Electro-Fotónica, Centro Universitario de Ciencias Exactas e Ingenierías, Universidad de Guadalajara, Blvd-M. García Barragán 1421, Guadalajara 44410, Jalisco, Mexico; antonio.czamora@academicos.udg.mx; 3Departamento de Ciencias Tecnológicas, Centro Universitario de la Ciénega (CUCienéga), Universidad de Guadalajara, Av. Universidad No. 1115, Lindavista, Ocotlán 47810, Jalisco, Mexico; eugenia.sanchez@academicos.udg.mx; 4Departamento de Ingeniería de Proyectos, Centro Universitario de Ciencias Exactas e Ingenierías, Universidad de Guadalajara, Blvd-M. García Barragán 1421, Guadalajara 44410, Jalisco, Mexico; hector.guillen1775@academicos.udg.mx; 5Departamento de Ciencias Básicas, Centro Universitario de la Ciénega (CUCienéga), Universidad de Guadalajara, Av. Universidad No. 1115, LindaVista, Ocotlán 47810, Jalisco, Mexico; maricela.jrodriguez@academicos.udg.mx

**Keywords:** low-fineness Fabry–Pérot interferometer, cavity length variations ΔL, pole-zero maps, bode plots, complex plane s

## Abstract

Pole-zero maps and Bode plots are commonly utilized in control systems and the study of natural phenomena to visualize their origins and behavior. In this paper, these graphical methods are applied to investigate the behavior of cavity variations, ΔL, in a low-finesse Fabry–Pérot interferometer subjected to external perturbations. Both graphical representations are analyzed in the s-plane. The study is theoretically performed, and the theory is corroborated by developing three numerical experiments where small displacements were applied. Based on the theoretical and numerical results, the cavity length variations, ΔL, can be studied on the s-plane applying the pole-zero maps and Bode plots. The two methods, including the theory and the experiments, are in agreement. Considering the theoretical and graphical results, pole-zero maps and Bode plots can be applied on the signal demodulation of optical interferometers and quasi-distributed sensors where local sensors are interferometers.

## 1. Introduction

An interferometer is applied in the measurement of physical parameters, such as temperature, pressure, stress, and voltage level, among others [1,2,3,4]. To obtain these measurements, the interference pattern needs to be demodulated. The interference pattern has the following shape: $I\left(\lambda ,\theta \right)=f_{e}\left(\lambda \right)f_{m}\left(\lambda ,\theta \right)$ [5,6,7] where $I\left(\lambda ,\theta \right)$ is the interference pattern according to the wavelength and phase produced by the distortion on the interferometer under study, $f_{e}\left(\lambda \right)$ is the envelope function of the wavelength, $f_{m}\left(\lambda ,\theta \right)$ is the modulated function, θ is the phase produced by a disturbance on the optical interferometer, and λ is the wavelength. Observing the interference pattern, if the modulated function $f_{m}\left(\lambda ,\theta \right)$ is filtered and the envelope function $f_{e}\left(\lambda \right)$ is eliminated through the filtering process, the data of the disturbance remains present within the modulated function $f_{m}\left(\lambda ,\theta \right)$. Given that the Laplace transform is a linear transformation, the information of the θ phase can be observed in the complex plane s.

The Laplace transformation is a mathematic transformation that performs a conversion of a function in the wavelength domain f(λ) into a complex function $F\left(s\right)$ [8,9], where $F\left(s\right)=\frac{M(s)}{N(s)}$ is a rational function: the polynomial numerator M(s) is employed to calculate the zeros system, and the denominator N(s) is deployed to calculate the positions of its poles. The coefficients of both depend on the characteristics of the system under observation.

For the analysis of dynamic systems, there are graphic methods that can be applied in the study of the physical system. Two of these methods are pole-zero maps and Bode plots. In pole-zero maps, a zero is represented by “O” and its positions are calculated by the roots of the polynomial $M\left(s\right)=0$. Meanwhile, a pole is represented by “X” and its positions are estimated by the roots of the polynomial $N\left(s\right)=0$. The positions of poles and zeros are employed to understand the behavior of the system under study [10,11]. Therefore, Bode plots are deployed to know the behavior of the magnitude and the function phase from the angular frequency. To implement the Bode plots, it was necessary to perform the following variable exchange: s=ωi and then the complex function $F\left(\omega i\right)$ is obtained. Following, the magnitude $\left|F\left(\omega i\right)\right|$ and phase $\phi \left(\omega i\right)$ are calculated from the complex function $F\left(\omega i\right)$. Once the magnitude and phase are known, the behavioral graphs of $\left|F\left(\omega i\right)\right|$ vs. ω and $\phi \left(\omega i\right)$ vs. ω are implemented in logarithmic form and, based on this analysis, the behavior or changes in the system-under-study can be documented [11,12,13].

According to the literature, graphical methods have not previously been employed for the study of cavity length changes caused by the perturbation on the interferometer cavity. In this paper, the cavity variations are graphically displayed employing pole-zero maps and Bode plots. For this study, the spectrum reflection of a low-fineness Fabry–Pérot interferometer $R\left(\lambda ,\Delta L\right)$ [5,6,7,9] is employed. When its cavity is disturbed by an external variable, its cavity length changes but the refraction index has very small variations; as a consequence, the variation in refraction index is not considered. From the interference pattern $R\left(\lambda ,\Delta L\right)$, the modulated function $f_{m}\left(\lambda ,\theta \left(\Delta L\right)\right)$ is filtered, causing the envelope function $f_{e}\left(\lambda \right)$ to be eliminated during the process. Applying the Laplace transform, the function $f_{m}\left(\lambda ,\theta \left(\Delta L\right)\right)$ is represented as the complex rational function $F_{m}\left(s,\theta \left(\Delta L\right)\right)=\frac{M(s,\theta \left(\Delta L\right))}{N(s)}$, and then zeros and poles are calculated. Afterwards, its positions are graphed in the pole-zero map, corroborating that the position of the zeros depend on the cavity length variations, ΔL, while the position of the poles is unaffected. Henceforth, on performing the variable exchange s=ωi, the modulated function $F_{m}\left(s,\theta \left(\Delta L\right)\right)$ is represented as $F_{m}\left(\omega ,\theta \left(\Delta L\right)\right)$; in this manner, the Bode plots are elaborated, obtaining behavioral graphs of $\left|F_{m}\left(\omega ,\theta \left(\Delta L\right)\right)\right|$ vs. ω and $\emptyset \left(\omega ,\theta \left(\Delta L\right)\right)$ vs. ω.

The theoretical and experimental results obtained in this paper are in agreement and conformity, the graphic methods can be applied in the analysis of cavity changes on the optical interferometers. Based on the results obtained, the pole-zero maps and Bode plots can be employed in the demodulation of interferograms.

## 2. Graphical Analysis of Complex Plane s Disturbance

### 2.1. Optical System

In references [9,10,11,13,14], the authors proposed an optical sensor based on two identical Bragg gratings, having a low reflectivity $\left(r\approx 1\%\right)$ with the goal of eliminating cross-talk noise. The interferometer was denominated low-fineness Fabry–Pérot due to its operating principle. This interferometer was deployed for the measurement of temperature, vibrations [6,15,16,17,18,19,20], and to study the behavior of perturbations in the complex plane s [21,22,23].

In references [24,25,26], a study on the Cross-Talk Noise was performed where the local sensors of a quasi-distributed sensor are Fabry-but interferometers were developed. In this study, it is demonstrated that the reflectivity of the Bragg grilles in the interferometer must be small to increase the number of local sensors and eliminate the Cross-Talk Noise. So, if the reflection coefficient is r=0.01 and using the fineness equation $F=\frac{\pi \sqrt{r}}{1-r}$ [26], The manufacture-but interferometer has a fineness of $F=\frac{\pi \sqrt{0.01}}{1-0.01}=0.317$.

Figure 1 presents the proposed optical system for the low-fineness Fabry–Pérot interferometer. The optical system is composed of a broadband light source, an optical spectrum analyzer (OSA spectrometer), a 50/50 optical circulator, and a monomodal optic fiber, which has a Fabry–Pérot interferometer based on two identical Bragg gratings. Both racks have low reflection to eliminate possible retro-reflections inside the interferometer linear cavity with a value of approximately r≈1% [5,6,7,14].

### 2.2. Optical Spectrum Due to an External Perturbation

Based on references [3,9], if the interferometer cavity is perturbed by an external variable and by applying signal processing, the interference pattern produced by the Fabry–Pérot interferometer takes the following shape:(1)$$\begin{array}{cc} R\left(\lambda ,\Delta L\right)=f_{e}\left(\lambda \right) & f_{m}\left(\lambda ,\Delta L\right)\\ & =2{\left[\frac{\pi nL_{BG}}{\lambda_{BG}}sinc\left(\frac{2nL_{BG}}{\lambda_{BG}^{2}}\lambda \right)\right]}^{2}\left[1+cos\left(\frac{4\pi nL}{\lambda_{BG}^{2}}\lambda -\frac{4\pi n\Delta L}{\lambda_{BG}}\right)\right] \end{array}$$
where $R\left(\lambda ,\Delta n\right)$ is the reflectance spectrum due to the disturbance, the enveloped function $f_{e}\left(\lambda \right)=2{\left[\frac{\pi nL_{BG}}{\lambda_{BG}}sinc\left(\frac{2nL_{BG}}{\lambda_{BG}^{2}}\lambda \right)\right]}^{2}$ is the reflection spectrum of the Bragg grating with a rectangular profile, π is the constant 3.1415…, $L_{BG}$ is the length of the Bragg grating, $\lambda_{BG}$ is the centered Bragg wavelength, n is the refractive index of the optic fiber, and λ is the wavelength. On the other hand, the modulated function $f_{m}\left(\lambda ,\Delta L\right)=1+cos\left(\frac{4\pi nL}{\lambda_{BG}^{2}}\lambda -\frac{4\pi n\Delta L}{\lambda_{BG}}\right)$ is the product of the interference caused between the reflections of both Bragg gratings: ΔL is the cavity length variation due to the external perturbation; L is the interferometer cavity (separation between the Bragg grating), and $\frac{4\pi n\Delta L}{\lambda_{BG}}$ is the phase produced by the cavity change ΔL when the interferometer cavity is disturbed. Therefore, to simplify the nomenclature of this paper, we introduce here the following definition:(2)$$\theta (\Delta L)=\frac{4\pi n\Delta L}{\lambda_{BG}}$$

In this case, the phase must satisfy the following inequality:(3)$$0\leq \theta \left(\Delta L\right)\leq 2\pi .$$

Substituting (2) in (1), the reflectance spectrum can be rewritten in the subsequent form:(4)$$R\left(\lambda ,\theta \left(\Delta L\right)\right)=2{\left[\frac{\pi nL_{BG}}{\lambda_{BG}}sinc\left(\frac{2nL_{BG}}{\lambda_{BG}^{2}}\lambda \right)\right]}^{2}\left[1+cos\left(\frac{4\pi nL}{\lambda_{BG}^{2}}\lambda -\theta \left(\Delta L\right)\right)\right]$$

Equation (4) describes the optical spectrum produced when the low-finesse Fabry-Pérot interferometer was perturbed by an external perturbation.

### 2.3. Complex Function $F_{m}\left(s,\theta \left(\Delta L\right)\right)$

When the Laplace transform is applied to a function, it is represented as a complex rational function $F\left(s\right)=\frac{M(s)}{N(s)}$ where M(s) is a polynomial employed to calculate the zeros, N(s) is a polynomial deployed to calculate the poles, and s=σ+ωi is a complex number where σ is real number, ω is the angular frequency, and i is a complex operator. This transformation can be applied for the study of perturbations over the interferometer since it is a linear transformation; data are not lost in the transformation process, and the information can be fully recovered through the inverse transformation. Additionally, observing the interferometer’s reflectivity (Equation (4)), the function $f_{e}\left(\lambda \right)$ contains no information regarding the $\theta \left(\Delta n\right)$ phase; nonetheless, the modulated function $f_{m}\left(\lambda ,\theta \left(\Delta L\right)\right)$ contains information about the $\theta \left(\Delta L\right)$ phase produced by the cavity length variations within the interferometer cavity.

It can be concluded that only the modulated function is required for the analysis of the interference pattern in the plane s [21,22]. Hence, the enveloped function can be removed through signal processing [4,7]. Hereafter, the problem is reduced to the following Laplace transform:(5)$$\begin{array}{cc} F_{m}\left(s,\theta \left(\Delta L\right)\right)=\int_{0}^{\infty } & f_{m}\left(\lambda ,\theta \left(\Delta L\right)\right)e^{-s\lambda }d\lambda \\ & =\int_{0}^{\infty }\left[1+cos\left(\frac{4\pi nL}{\lambda_{BG}^{2}}\lambda -\theta \left(\Delta L\right)\right)\right]e^{-s\lambda }d\lambda , \end{array}$$

Through linearity property, we obtain the following:(6)$$F_{m}\left(s,\theta \left(\Delta L\right)\right)=\int_{0}^{\infty }e^{-s\lambda }d\lambda +\int_{0}^{\infty }cos\left(\frac{4\pi nL}{\lambda_{BG}^{2}}\lambda -\theta \left(\Delta L\right)\right)e^{-s\lambda }d\lambda .$$

It can be noted that integral (6) entertains no dependency on the $\theta \left(\Delta L\right)$ phase; consequently, it does not depend on cavity length variations, ΔL, and, subsequently, its solution is as follows:(7)$$F_{m}\left(s,\theta \left(\Delta L\right)\right)=\frac{1}{s}+\frac{cos\left[\theta \left(\Delta L\right)\right]s}{s^{2}+{\left(\frac{4\pi nL}{\lambda_{BG}^{2}}\right)}^{2}}+\frac{\frac{4\pi nL}{\lambda_{BG}^{2}}sin\left[\theta \left(\Delta L\right)\right]}{s^{2}+{\left(\frac{4\pi nL}{\lambda_{BG}^{2}}\right)}^{2}}.$$

On developing (7), the next expression is reached:(8)$$\begin{array}{cc} F_{m}\left(s,\theta \left(\Delta L\right)\right) & =\frac{M(s,\theta \left(\Delta L\right))}{N(s)}\\ & =\frac{\left(1+cos\left[\theta \left(\Delta L\right)\right]\right)s^{2}+\frac{4\pi nL}{\lambda_{BG}^{2}}sin\left[\theta \left(\Delta L\right)\right]s+{\left(\frac{4\pi nL}{\lambda_{BG}^{2}}\right)}^{2}}{s\left[s^{2}+{\left(\frac{4\pi nL}{\lambda_{BG}^{2}}\right)}^{2}\right]}\;. \end{array}$$

Onward from (8), the polynomial $M(s,\theta \left(\Delta L\right))$ is required to calculate the zeros of the complex function:(9)$$M(s,\theta \left(\Delta L\right))=\left(1+cos\left[\theta \left(\Delta L\right)\right]\right)s^{2}+\frac{4\pi nL}{\lambda_{BG}^{2}}sin\left[\theta \left(\Delta L\right)\right]s+{\left(\frac{4\pi nL}{\lambda_{BG}^{2}}\right)}^{2}=0$$
and polynomial N(s) is required to calculate its poles:(10)$$N\left(s\right)=s\left[s^{2}+{\left(\frac{4\pi nL}{\lambda_{BG}^{2}}\right)}^{2}\right]=0.$$

### 2.4. Graphic Representation of the Phase $\theta \left(\Delta L\right)$

When the Fabry–Pérot interferometer cavity is disturbed, the cavity modified and as a consequence, the optical path difference is also changed, the interference pattern has a phase $\theta \left(\Delta L\right)$, as indicated in Equations (2) and (4). This phase can be observed in the complex function $F_{m}\left(s,\theta \left(\Delta L\right)\right)$ (Equation (8)) and in the polynomial $M\left(s,\theta \left(\Delta L\right)\right)$ (Equation (9)).

Based on Equation (9), the position of Zero is according to the cavity length variation, and it can be observed graphically in the pole-zero map. The plane s will be employed to represent the complex function $F_{m}\left(s,\theta \left(\Delta L\right)\right)$ graphically where the imaginary number s=σ+ωi is a real part and an imaginary one: σ is a real number; ω is the angular frequency, and i is the complex operator as was mentioned. The horizontal axis corresponds to the real values σ while the vertical axis belongs to the imaginary values ωi.

Mode plots provide an alternative graphical method for studying the behavior of dynamic systems. This representation illustrates the magnitude and phase response as a function of frequency. The magnitude is presented on a logarithmic scale and expressed in decibels. It is applied in Section 2.4.2.

#### 2.4.1. Pole-Zero Map

The pole-zero map is an alternative for performing the dynamic analysis of disturbances regarding the cavity of an optical interferometer. On the map, a zero is presented by an “O” and a pole is represented by an “X” [5,9].

To calculate the poles of the function $F_{p}\left(s,\theta \left(\Delta n\right)\right)$, the polynomial indicated by Equation (10) must be resolved:(11)$$N\left(s\right)=s\left[s^{2}+{\left(\frac{4\pi nL}{\lambda_{BG}^{2}}\right)}^{2}\right]=0\;\;\to \begin{array}{c} s_{0}=0\\ s_{1}=\frac{4\pi nL}{\lambda_{BG}^{2}}i\\ s_{2}=-\frac{4\pi nL}{\lambda_{BG}^{2}}i \end{array}\;\;\;.$$

From Equations (10) and (11), the complex function $F_{p}\left(s,\theta \left(\Delta L\right)\right)$ contains three poles with its position depending on the refractive index of the optic fiber n, the cavity length of the Fabry–Pérot interferometer L, and the Bragg wavelength $\lambda_{BG}$. The pole $s_{0}$ is positioned in the origin of plane s, pole $s_{1}$ is on the positive side of the imaginary axis, and pole $s_{2}$ is found in the negative space of the imaginary axis. Furthermore, to calculate the zeros of the complex function, Equation (9) must be resolved, the latter able to be rewritten in the following form:(12)$$M\left(s\right)=s^{2}+\frac{4\pi nLsin\left[\theta \left(\Delta L\right)\right]}{\lambda_{BG}^{2}\left(1+cos\left[\theta \left(\Delta L\right)\right]\right)}s+\frac{{\left(\frac{4\pi nL}{\lambda_{BG}^{2}}\right)}^{2}}{\left(1+cos\left[\theta \left(\Delta L\right)\right]\right)}=0.$$

If the perfect squared binomial is considered for the solution of the squared Equation (12), we express the equation in the following manner:(13)$$s^{2}+\frac{4\pi nLsin\left[\theta \left(\Delta L\right)\right]}{\lambda_{BG}^{2}\left(1+cos\left[\theta \left(\Delta L\right)\right]\right)}s+{\left[\frac{4\pi nLsin\left[\theta \left(\Delta L\right)\right]}{2\lambda_{BG}^{2}\left(1+cos\left[\theta \left(\Delta L\right)\right]\right)}\right]}^{2}-{\left[\frac{4\pi nLsin\left[\theta \left(\Delta L\right)\right]}{2\lambda_{BG}^{2}\left(1+cos\left[\theta \left(\Delta L\right)\right]\right)}\right]}^{2}=-\frac{{\left(\frac{4\pi nL}{\lambda_{BG}^{2}}\right)}^{2}}{\left(1+cos\left[\theta \left(\Delta L\right)\right]\right)}.$$

From Equation (13), we obtain the following:(14)$$s^{2}+\frac{4\pi nLsin\left[\theta \left(\Delta L\right)\right]}{\lambda_{BG}^{2}\left(1+cos\left[\theta \left(\Delta L\right)\right]\right)}s+{\left[\frac{2\pi nLsin\left[\theta \left(\Delta L\right)\right]}{\lambda_{BG}^{2}\left(1+cos\left[\theta \left(\Delta L\right)\right]\right)}\right]}^{2}=-\frac{{\left(\frac{4\pi nL}{\lambda_{BG}^{2}}\right)}^{2}}{\left(1+cos\left[\theta \left(\Delta L\right)\right]\right)}+{\left[\frac{2\pi nLsin\left[\theta \left(\Delta L\right)\right]}{\lambda_{BG}^{2}\left(1+cos\left[\theta \left(\Delta L\right)\right]\right)}\right]}^{2}.$$

Completing the perfect squared binomial, we reach the following:(15)$${\left[s+\frac{2\pi nLsin\left[\theta \left(\Delta L\right)\right]}{\lambda_{BG}^{2}\left(1+cos\left[\theta \left(\Delta L\right)\right]\right)}\right]}^{2}={\left[\frac{2\pi nLsin\left[\theta \left(\Delta L\right)\right]}{\lambda_{BG}^{2}\left(1+cos\left[\theta \left(\Delta L\right)\right]\right)}\right]}^{2}-\frac{{\left(\frac{4\pi nL}{\lambda_{BG}^{2}}\right)}^{2}}{\left(1+cos\left[\theta \left(\Delta L\right)\right]\right)}\;.$$

On simplifying the term on the right in the equality, the following is obtained:(16)$${\left[s+\frac{2\pi nLsin\left[\theta \left(\Delta L\right)\right]}{\lambda_{BG}^{2}\left(1+cos\left[\theta \left(\Delta L\right)\right]\right)}\right]}^{2}=\frac{4{\left(\pi nL\right)}^{2}}{\lambda_{BG}^{4}\left(1+cos\left[\theta \left(\Delta L\right)\right]\right)}\left[\frac{{sin}^{2}\left[\theta \left(\Delta L\right)\right]}{1+cos\left[\theta \left(\Delta L\right)\right]}-4\right]\;.$$

To conclude, the position of the zeros is given by the following:(17)$$s_{0,1}=-\frac{2\pi nLsin\left[\theta \left(\Delta L\right)\right]}{\lambda_{BG}^{2}\left(1+cos\left[\theta \left(\Delta L\right)\right]\right)}\pm \sqrt{\frac{{4\left(\pi nL\right)}^{2}}{\lambda_{BG}^{4}\left(1+cos\left[\theta \left(\Delta L\right)\right]\right)}\left[\frac{{sin}^{2}\left[\theta \left(\Delta L\right)\right]}{1+cos\left[\theta \left(\Delta L\right)\right]}-4\right]}.$$

If the last expression is reduced, we can determine the following:(18)$$s_{0,1}=-\frac{2\pi nL}{\lambda_{BG}^{2}}\frac{sin\left[\theta \left(\Delta L\right)\right]}{1+cos\left[\theta \left(\Delta L\right)\right]}\pm \frac{2\pi nL}{\lambda_{BG}^{2}}\sqrt{\frac{1}{\left(1+cos\left[\theta \left(\Delta L\right)\right]\right)}\left[\frac{{sin}^{2}\left[\theta \left(\Delta L\right)\right]}{1+cos\left[\theta \left(\Delta L\right)\right]}-4\right]}.$$

Based on expression (18), the position of the zeros is in accordance with the physical parameters of the interferometer and the cavity length variation. In Figure 2a, the pole-zero map is illustrated when the interferometer is perturbed by an external variable and the phase has its value of $\theta \left(\Delta L\right)=50{}^{\circ}$. Meanwhile, Figure 2b presents the pole-zero map when the cavity length variation produces the phases $\theta \left(\Delta L\right)$: 0°; 40°; 80°; 120°; 240°; 280°, and 320°. For both figures, the parameters were n=1.46, $\lambda_{BG}=1534.5\;nm$, and L=5 mm.

Based on Figure 2 and the pole-zero map, the phase $\theta \left(\Delta L\right)$ is represented by a complex number that can withhold a real term and an imaginary one. Thus, the following points are noteworthy:
The poles always maintain the same positions: $s_{0}=0$, $s_{1}=\frac{4\pi nL}{\lambda_{BG}^{2}}i$, $s_{2}=-\frac{4\pi nL}{\lambda_{BG}^{2}}i$.When $\theta \left(\Delta L\right)=0{}^{\circ}$, the zeros have positions $s_{0}=\frac{2\sqrt{2}\pi nL}{\lambda_{BG}^{2}}i$, $s_{1}=-\frac{2\sqrt{2}\pi nL}{\lambda_{BG}^{2}}i$ and when they are placed over the imaginary axis of plane s (see Figure 2a). In this case, it is implied that the Fabry–Pérot interferometer is not disturbed by an external variable and that it performs as a critically stable system.

**Proof.** Taking Equation (18) and on knowing $\theta \left(\Delta L\right)=0{}^{\circ}$, we obtain the following: (19)$$s_{0,1}=\frac{2\pi nL}{\lambda_{BG}^{2}}\left\{-\frac{sin\left(0\right)}{1+cos\left(0\right)}\pm \sqrt{\frac{1}{\left(1+cos\left(0\right)\right)}\left[\frac{{sin}^{2}\left(0\right)}{1+cos\left(0\right)}-4\right]}\right\}.$$On evaluation, $-\frac{sin\left(0\right)}{1+cos\left(0\right)}=0$ and $\frac{{sin}^{2}\left(0\right)}{1+cos\left(0\right)}=0$ is fulfilled, giving as a result the following:(20)$$\begin{array}{c} s_{0}=\frac{2\sqrt{2}\pi nL}{\lambda_{BG}^{2}}i\\ s_{1}=-\frac{2\sqrt{2}\pi nL}{\lambda_{BG}^{2}}i \end{array}$$□


3.When $\theta \left(\Delta L\right)=360{}^{\circ}$, the zeros have the positions $s_{0}=\frac{2\sqrt{2}\pi nL}{\lambda_{BG}^{2}}i$ and $s_{1}=-\frac{2\sqrt{2}\pi nL}{\lambda_{BG}^{2}}i$, and they are localized on the imaginary axis of plane s. In this case, the disturbance leads to a full phase cycle.4.When $\theta \left(\Delta L\right)=180{}^{\circ}$, the positions of the zeros $s_{0,1}$ is undetermined.


**Proof.** On taking Equation (18) and on knowing $\theta \left(\Delta L\right)=180{}^{\circ}$, we obtain: (21)$$s_{0,1}=\frac{2\pi nL}{\lambda_{BG}^{2}}\left\{-\frac{sin\left(180\right)}{1+cos\left(180\right)}\pm \sqrt{\frac{1}{\left(1+cos\left(180\right)\right)}\left[\frac{{sin}^{2}\left(180\right)}{1+cos\left(180\right)}-4\right]}\right\}.$$On knowing $sin\left(180\right)=0$ and $cos\left(180\right)=-1$, we reach the following:(22)$$s_{0,1}=\frac{2\pi nL}{\lambda_{BG}^{2}}\left\{-\frac{0}{1-1}\pm \sqrt{\frac{1}{\left(1-1\right)}\left[\frac{0}{1-1}-4\right]}\right\}.$$From Equation (22), the position of the zeros remains undetermined by the term $\frac{-4}{0}$.
□


5.When $\theta \left(\Delta L\right)\neq 0{}^{\circ},\;\theta \left(\Delta L\right)\neq 180{}^{\circ}$ and $\theta \left(\Delta L\right)=360{}^{\circ}$, the position of the zeros comprises a complex number that has a real term and an imaginary one (see Figure 2b). This is true because the next condition is satisfied.



(23)
$$\frac{{sin}^{2}\left[\theta \left(\Delta n\right)\right]}{1+cos\left[\theta \left(\Delta n\right)\right]}<4,$$


Consequently, $\sqrt{\frac{1}{\left(1+cos\left[\theta \left(\Delta n\right)\right]\right)}\left[\frac{{sin}^{2}\left[\theta \left(\Delta n\right)\right]}{1+cos\left[\theta \left(\Delta n\right)\right]}-4\right]}$ generates a complex number.

#### 2.4.2. Bode Plots

In this case, the complex function $F_{m}\left(s,\theta \left(\Delta L\right)\right)$ is analyzed with the goal of knowing the behavior of the phase $\theta \left(\Delta L\right)$. To achieve this, the change in variable s=ωi is employed, whereupon Expression (8) shifts to the following shape:(24)$$F_{m}\left(\omega i,\theta \left(\Delta L\right)\right)=\frac{\left(1+cos\left[\theta \left(\Delta L\right)\right]\right){\left(\omega i\right)}^{2}+\omega \frac{4\pi nL}{\lambda_{BG}^{2}}sin\left[\theta \left(\Delta n\right)\right]i+{\left(\frac{4\pi nL}{\lambda_{BG}^{2}}\right)}^{2}}{\left[{\left(i\omega \right)}^{3}+\omega {\left(\frac{4\pi nL}{\lambda_{BG}^{2}}\right)}^{2}i\right]}\;.$$

On knowing $i^{2}=-1$ and taking Equation (24) into consideration, we have the following:(25)$$F_{m}\left(\omega i,\theta \left(\Delta L\right)\right)=\frac{-\left(1+cos\left[\theta \left(\Delta L\right)\right]\right)\omega^{2}+{\left(\frac{4\pi nL}{\lambda_{BG}^{2}}\right)}^{2}+\omega \frac{4\pi nL}{\lambda_{BG}^{2}}sin\left[\theta \left(\Delta L\right)\right]i}{\left[-\omega^{3}+\omega {\left(\frac{4\pi nL}{\lambda_{BG}^{2}}\right)}^{2}\right]i}.$$

On separating real terms from imaginary ones in Expression (25), we determine the following:(26)$$F_{m}\left(\omega i,\theta \left(\Delta L\right)\right)=\frac{\omega \frac{4\pi nL}{\lambda_{BG}^{2}}sin\left[\theta \left(\Delta L\right)\right]}{-\omega^{3}+\omega {\left(\frac{4\pi nL}{\lambda_{BG}^{2}}\right)}^{2}}+\frac{\left(1+cos\left[\theta \left(\Delta L\right)\right]\right)\omega^{2}-{\left(\frac{4\pi nL}{\lambda_{BG}^{2}}\right)}^{2}}{-\omega^{3}+\omega {\left(\frac{4\pi nL}{\lambda_{BG}^{2}}\right)}^{2}}i.$$

The complex function $F_{m}\left(\omega i,\theta \left(\Delta L\right)\right)$ has its magnitude defined by the following:(27)$$\left|F_{m}\left(\omega i,\theta \left(\Delta L\right)\right)\right|=\sqrt{{\left[\frac{\omega \frac{4\pi nL}{\lambda_{BG}^{2}}sin\left(\theta \left(\Delta L\right)\right)}{-\omega^{3}+\omega {\left(\frac{4\pi nL}{\lambda_{BG}^{2}}\right)}^{2}}\right]}^{2}+{\left[\frac{\left(1+cos\left(\theta \left(\Delta L\right)\right)\right)\omega^{2}-{\left(\frac{4\pi nL}{\lambda_{BG}^{2}}\right)}^{2}}{-\omega^{3}+\omega {\left(\frac{4\pi nL}{\lambda_{BG}^{2}}\right)}^{2}}\right]}^{2}}.$$
and its phase $\emptyset \left(\omega i,\theta \left(\Delta L\right)\right)$ is given by the following:(28)$$\emptyset \left(\omega i,\theta \left(\Delta L\right)\right)={tg}^{-1}\left(\frac{\left(1+cos\left(\theta \left(\Delta L\right)\right)\right)\omega^{2}-{\left(\frac{4\pi nL}{\lambda_{BG}^{2}}\right)}^{2}}{\omega \frac{4\pi nL}{\lambda_{BG}^{2}}sin\left(\theta \left(\Delta L\right)\right)}\right).$$
where $\left|F_{m}\left(\omega i,\theta \left(\Delta L\right)\right)\right|$ is the magnitude of the complex function $F_{m}\left(\omega i,\theta \left(\Delta L\right)\right),$ and $\emptyset \left(\omega i,\theta \left(\Delta L\right)\right)$ is the phase. Both $\left|F_{m}\left(\omega i,\theta \left(\Delta L\right)\right)\right|$ and $\emptyset \left(\omega i,\theta \left(\Delta L\right)\right)$ are based on the angular frequency and the perturbance parameter on the interferometer. Knowing that the magnitude and phase can be estimated through the complex function, Bode plots are implemented considering the following expression [11,12,13]:(29)$${\left|F_{m}\left(\omega i,\theta \left(\Delta L\right)\right)\right|}_{Db}=20{log}_{10}\left(\left|F_{p}\left(\omega i,\theta \left(\Delta L\right)\right)\right|\right).$$

Therefore, Bode plots are presented as a graph for the magnitude vs. the angular- frequency logarithm in decibels, and as another graph for phase vs. the angular-frequency logarithm. Figure 3a presents a Bode diagram when the Fabry–Pérot interferometer is perturbed, and the phase has the value of $\theta \left(\Delta L\right)=50^{{}^{\circ}}$. Meanwhile, Figure 3b displays a Bode diagram when the interferometer undergoes different external perturbations, and as a consequence, the value phase $\theta \left(\Delta L\right)$ changes: 0°, 40°, 80°, 120°, 160°, 200°, 240°, 280°, 320°, and 360°. The parameters were n=1.46, $\lambda_{BG}=1534.5\;nm$, and L=5 mm.

Observing Figure 3, Bode diagrams show the magnitude behavior vs. frequency and phase vs. frequency. Figure 3a shows the winery diagrams for Equations (27) and (28) where the parameters are n=1.46, $\lambda_{BG}=1534.5\;nm$ and L=5 mm and $\theta \left(\Delta L\right)=50^{{}^{\circ}}$. On the other hand, Figure 3b shows the winery diagrams for the same Equations (27) and (28) with n=1.46, $\lambda_{BG}=1534.5\;nm$ and L=5 mm but phase $\theta \left(\Delta L\right)$ has a value of $0^{{}^{\circ}}$,$\;40^{{}^{\circ}}$, 80°, 120°, 160°, 200°, 240°,  280°, 320° and 360°. Analyzing Figure 3b, the disturbances in the interferometer can be detected through the behavior graph of phase vs. frequency since small phase variation $\theta \left(\Delta L\right)$ They are visualized in the aforementioned behavior graph. On the other hand, the disturbance cannot be visualized in the magnitude graph vs. frequency because phase information is loss when calculating the magnitude of the complex function.

## 3. Numerical Experiments

Three numerical experiments were developed to demonstrate the proposed theory on the graphic representation of cavity length variation in the complex plane s employing the pole-zero map, Bode plots, and a low-fineness Fabry–Pérot interferometer. In the three experiments, the interference pattern in the Fabry–Pérot interferometer $R\left(\lambda ,\theta \left(\Delta L\right)\right)$ is simulated and the parameters are similar to those employed in reference [5,15]: n=1.46, $L_{BG}=1\;mm$; $\lambda_{BG}=1534\;nm$; L=5 mm, and λ is within the interval of −1 until 1 nm, with a sampling rate of Z = 1024, and each experiment is performed with a different value of refractive index variations.

In each experiment, the modulated function $f_{m}\left(\lambda ,\theta \left(\Delta L\right)\right)$ is filtered. Therefore, the complex rational function $F_{m}\left(s,\theta \left(\Delta L\right)\right)$ is calculated through a Laplace transform. Subsequently, the pole-zero maps and the Bode plots are devised for the function $F_{m}\left(s,\theta \left(\Delta L\right)\right)$. Based on the parameters, we obtain: $\frac{4\pi nL}{\lambda_{BG}^{2}}=38.983\;\left[\frac{cycle}{nm}\right];\;\frac{2nL_{BG}}{\lambda_{BG}^{2}}=1.24\;nm,$ and $\frac{\pi nL_{BG}}{\lambda_{BG}}=2.99\times 10^{3}$. Numerical simulations were performed by means of MATLAB R2023a scientific software.

### 3.1. Experiment 1: ΔL=0 μm and $\theta \left(\Delta L\right)=0^{{}^{\circ}}$

When the interferometer is unperturbed by any external source, the cavity length variation is zero ΔL=0; consequently, the phase $\theta \left(\Delta L\right)$ also has a value of zero. Therefore, the resulting interference pattern in the Fabry–Pérot interferometer has the shape:(30)$${\left.R\left(\lambda ,\theta \left(\Delta L\right)\right)\right\rfloor }_{\theta \left(\Delta L\right)=0{}^{\circ}}=2{\left[2.99\times 10^{3}sinc\left(1.24\lambda \right)\right]}^{2}\left[1+cos\left(38.983\lambda \right)\right].$$

Developing the signal processing proposed in references [5], the modulated function is filtered, obtaining:(31)$$f_{m}{\left.\left(\lambda ,\theta \left(\Delta L\right)\right)\right|}_{\theta \left(\Delta L\right)=0{}^{\circ}}=1+cos\left(38.983\lambda \right)$$

In Figure 4a. the interference pattern is illustrated and Figure 4b displays the modulated function $f_{m}{\left.\left(\lambda ,\theta \left(\Delta L\right)\right)\right|}_{\theta \left(\Delta L\right)=0{}^{\circ}}$. Employing the Laplace transform, the complex function ${\left.F_{p}\left(s,\theta (\Delta L\right)\right|}_{\theta \left(\Delta L\right)=0{}^{\circ}}$ is determined as follows:(32)$${\left.F_{m}\left(s,\theta \left(\Delta L\right)\right)\right|}_{\theta \left(\Delta L\right)=0{}^{\circ}}=\frac{2s^{2}+1519.674}{s\left[s^{2}+1519.674\right]},$$

To illustrate Equation (32), its pole-zero map is presented in Figure 4c, and its Bode diagram is displayed in Figure 4d.

**Figure 4 sensors-25-02182-f004:**
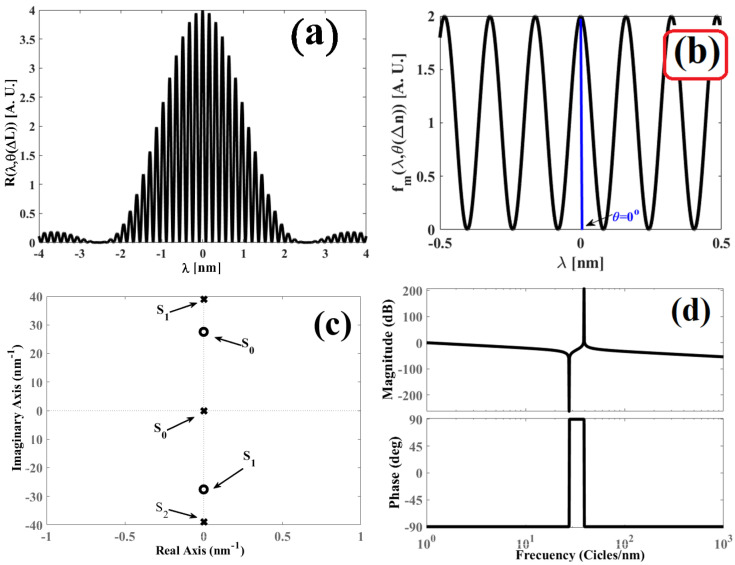
Obtained graphs for the refractive index variation ΔL=0, $\left(\theta (\Delta L)=0\right)$: (**a**) Interference pattern $R\left(\lambda ,\theta (\Delta L)\right)$ produced by the Fabry–Pérot interferometer; (**b**) modulated function $f_{m}{\left.\left(\lambda ,\theta \left(\Delta L\right)\right)\right|}_{\theta \left(\Delta L\right)=0^{{}^{\circ}}}$ filtered from the interference pattern $R\left(\lambda ,\theta (\Delta L)\right)$; (**c**) pole-zero map obtained for the function ${\left.F_{m}\left(s,\theta \left(\Delta L\right)\right)\right|}_{\theta \left(\Delta L\right)=0^{{}^{\circ}}}$, and (**d**) Bode diagram obtained for the function ${\left.F_{m}\left(s,\theta \left(\Delta L\right)\right)\right|}_{\theta \left(\Delta L\right)=0^{{}^{\circ}}}$.

Based on Figure 4a,b, the interferometer has no external disturbance, as a consequence, the interference pattern ${\left.R\left(\lambda ,\theta \left(\Delta L\right)\right)\right\rfloor }_{\theta \left(\Delta L\right)=0{}^{\circ}}$ and the modulated function $f_{m}{\left.\left(\lambda ,\theta \left(\Delta L\right)\right)\right|}_{\theta \left(\Delta L\right)=0{}^{\circ}}$ They are in phase. In the pole-zero map indicated by Figure 4c, when the cavity of the interferometer is undisturbed, the cavity length variation is ΔL=0 and the phase is also $\theta \left(\Delta L\right)=0^{{}^{\circ}}$; consequently, the position of the zeros and the poles are located over the imaginary axis. In this case, the positions of the zeros are $s_{0}=27.565i$ and $s_{1}=-27.565i$, while the positions of the poles are $s_{0}=0$, $s_{1}=38.983i$ and $s_{2}=-38.983i$. In the Bode diagram shown in Figure 4d, when the interferometer is unperturbed $\left(\theta \left(\Delta L\right)=0{}^{\circ}\right)$, the phase ${\left.\emptyset \left(\omega i,\theta \left(\Delta L\right)\right)\right|}_{\theta \left(\Delta L\right)=0}$ has a value of −90°, and this suddenly changes to 90°; in this transition, the magnitude ${\left|F_{m}\left(\omega i,\theta \left(\Delta L\right)\right)\right|}_{\theta \left(\Delta L\right)=0\;}$ has a negative asymptotic. Afterwards, the phase ${\left.\emptyset \left(\omega i,\theta \left(\Delta L\right)\right)\right|}_{\theta \left(\Delta L\right)=0}$ maintains its constant value of 90° for a frequency interval, while the magnitude ${\left|F_{m}\left(\omega i,\theta \left(\Delta L\right)\right)\right|}_{\theta \left(\Delta L\right)=0\;}$ encounters small variations. When the phase ${\left.\emptyset \left(\omega i,\theta \left(\Delta L\right)\right)\right|}_{\theta \left(\Delta L\right)=0}$ undergoes a transition from 90° to –90°, the magnitude ${\left|F_{m}\left(\omega i,\theta \left(\Delta L\right)\right)\right|}_{\theta \left(\Delta L\right)=0\;}$ has a positive asymptotic. To conclude, the phase remains constant at –90°, and the magnitude presents small variations. However, when the interferometer cavity is disturbed by an external force $\left(\theta \left(\Delta L\right)\neq 0\right)$, there are changes in the phase ${\left.\emptyset \left(\omega i,\theta \left(\Delta L\right)\right)\right|}_{\theta \left(\Delta L\right)\neq 0}$. These variations can be observed in the Bode diagram as displayed in Figure 3b and Figure 4d. Based on our results, the theoretical results and experimental results are in agreement. This can be observed in the behavior graphs of $\left|F_{m}\left(\omega i,\theta \left(\Delta L\right)\right)\right|$ vs. ω and $\emptyset \left(\omega i,\theta \left(\Delta L\right)\right)$ vs. ω.

### 3.2. Experiment 2: ΔL=102.08 nm and $\theta \left(\Delta L\right)=70^{{}^{\circ}}$

When the interferometer cavity is disturbed and has a cavity length variation of ΔL=157.0129 μm, the phase is $\theta \left(\Delta n\right)=70^{{}^{\circ}}$; then, the interference pattern takes the following shape:(33)$${\left.R\left(\lambda ,\theta \left(\Delta n\right)\right)\right\rfloor }_{\theta \left(\Delta n\right)=70{}^{\circ}}=2{\left[2.99\times 10^{3}sinc\left(1.24\lambda \right)\right]}^{2}\left[1+cos\left(38.983\lambda -70\right)\right]$$

If the function ${\left.f_{m}\left(\lambda ,\theta \left(\Delta L\right)\right)\right|}_{\theta \left(\Delta L\right)=70{}^{\circ}}$ is filtered from the interferometer reflectance spectrum [5], the resulting function will be the following:(34)$${\left.f_{m}\left(\lambda ,\theta \left(\Delta L\right)\right)\right|}_{\theta \left(\Delta L\right)=70{}^{\circ}}=1+cos\left(38.983\lambda -70\right).$$

Meanwhile, its complex function is as follows:(35)$${\left.F_{m}\left(s,\theta \left(\Delta L\right)\right)\right|}_{\theta \left(\Delta L\right)=70{}^{\circ}}=\frac{1.633s^{2}+30.17s+1519.674}{s\left[s^{2}+1519.674\right]}.$$

In Figure 5a, the interference pattern of the interferometer is illustrated, Figure 5b displays the function ${\left.f_{m}\left(\lambda ,\theta \left(\Delta L\right)\right)\right|}_{\theta \left(\Delta L\right)=70{}^{\circ}}$ with the displacement of $70^{{}^{\circ}}$, and Figure 5c,d present the pole-zero map and the Bode diagram of the complex function (32).

Comparing the theory and 5c, theory and experiment are in agreement. In the pole-zero map, if the cavity length variation is ΔL=102.08 nm, then the phase has a value of $\theta \left(\Delta L\right)=70^{{}^{\circ}}$; therefore, the zeros are displaced out of the complex axis, taking the following real and imaginary values: $s_{0}=-9.24+29.1i$ and $s_{1}=-9.24-29.1i$. Meanwhile, the poles maintain the same positions as follows: $s_{0}=0$, $s_{1}=-38.983i$ and $s_{2}=-38.983i$. Furthermore, comparing Figure 3b and Figure 5d, the Bode plots are in agreement, corroborating the theory based on the numerical experiments. Clearly, the behavior of graphs $\left|F_{m}\left(\omega ,\theta \left(\Delta L\right)\right)\right|$ vs. ω and $\emptyset \left(\omega ,\theta \left(\Delta L\right)\right)$ vs. ω change when the cavity of the Fabry–Pérot interferometer is disturbed.

### 3.3. Experiment 3: ΔL=204.26 nm and $\theta \left(\Delta L\right)=140^{{}^{\circ}}$

Moreover, if the cavity of the Fabry–Pérot interferometer is perturbed and if ΔL=314.205 μm, then its face has a value of $\theta \left(\Delta L\right)=140^{{}^{\circ}}$ and, consequently, the interference pattern holds the following shape:(36)$${\left.R\left(\lambda ,\theta \left(\Delta L\right)\right)\right\rfloor }_{\theta \left(\Delta L\right)=140{}^{\circ}}=2{\left[2.99\times 10^{3}sinc\left(1.24\lambda \right)\right]}^{2}\left[1+cos\left(38.983\lambda -140\right).\right]$$

If the function $f_{m}\left(\lambda ,\theta \left(\Delta L\right)\right)$ is filtered from the interference pattern [5], we obtain(37)$${\left.f_{m}\left(\lambda ,\theta \left(\Delta L\right)\right)\right\rfloor }_{\theta \left(\Delta L\right)=140{}^{\circ}}=1+cos\left(38.983\lambda -140\right).$$

On applying the Laplace transform, the following complex function is reached:(38)$${\left.F_{m}\left(s,\theta \left(\Delta L\right)\right)\right\rfloor }_{\theta \left(\Delta L\right)=140{}^{\circ}}=\frac{0.8022s^{2}+38.2s+1519.674}{s\left[s^{2}+1519.674\right]}.$$

In Figure 6a, the interference pattern is presented, Figure 6b provides the function ${\left.f_{m}\left(\lambda ,\theta \left(\Delta L\right)\right)\right\rfloor }_{\theta \left(\Delta L\right)=140{}^{\circ}}$, and in Figure 6c,d, the pole-zero map and the Bode diagram are displayed.

Conducting a comparison between 2a and 6c, the zeros are outside of the complex axis due to the cavity length variation ΔL=204.26 nm $\left(\theta \left(\Delta L\right)=140{}^{\circ}\right)$, and the positions are $s_{0}=-23.8+36.4i$ and $s_{1}=-23.8-36.4i$. Meanwhile, the poles retain their positions: $s_{0}=0$, $s_{1}=-38.983i$ and $s_{2}=-38.983i$. Once again, the results corroborate the theory proposed in the previous sections. Comparing the diagrams displayed in Figure 3b and Figure 6d, both graphs present nearly the exact same behavior. Henceforth, theory and experiment are in agreement.

### 3.4. Comparison

After conducting the numerical experiments, a comparison of the graphical results is presented in this section. By examining the interference patterns shown in Figure 4a, Figure 5a and Figure 6a, it is evident that their shapes are very similar. This similarity arises because the perturbations are small, leading to minor changes in the cavity length. As a consequence, the phase satisfies the condition specified by Equation (3).

On the other hand, Table 1 provides a comparative analysis of the graphs displayed in Figure 4b–d, Figure 5b–d and Figure 6b–d, which correspond to the experimental results.

Describing Table 1, in first column, the variation in cavity length is shown due to the disturbance on the optical interferometer. In the second column, the phase generated by the change in length of cavity is shown, and the position of the poles and zeros is shown in the column 3 phase. From Table 1, you have the following comparisons:

First: the results correspond to the cases: (a) interferometer has no external disturbance $\left(\Delta L=0nm\to \theta \left(\Delta L\right)=0{}^{\circ}\right)$, and this can be observed in line 1; (b) interferometer is disturbed, and these can be observed in lines 2 and 3 where $\Delta L=102.08\;nm\to \theta \left(\Delta L\right)=70{}^{\circ}$ and $\Delta L=204.26\;nm\to \theta \left(\Delta L\right)=140{}^{\circ}$, respectively.

Second: the variation in cavity length is observable in the modulated function, as it produces a phase shift proportional to the magnitude of the variable. In Figure 4b, the modulated function is in phase because there is no perturbation in the optical interferometer, whereas in Figure 5b and Figure 6b, the function is out of phase due to the perturbation of the interferometer by an external variable.

Third: the poles maintain their position regardless of whether the interferometer is perturbed or not, as shown in Table 1, section “Pole-Zero Map”, column 1.

Fourth: the position of the zeros depends on the value of the phase produced by the distortion on the optical interferometer. In Figure 4c, the zeros are located on the imaginary axis because the interferometer is not perturbed. However, in Figure 5c and Figure 6c, the zeros are located with both real and imaginary terms (they are off the imaginary axis) due to the external perturbation on the interferometer, as shown in Table 1, section “Pole-Zero Map”, column 2.

Fifth: the number of asymptotes and phase changes depends on the perturbation of the interferometer. If the interferometer is unperturbed, there are two asymptotes and two phase changes (see Figure 4d). However, there is only one asymptote and one phase change when the interferometer cavity is perturbed by an external variable.

Based on these comparisons, the phase change in the interference pattern is observable in a pole-zero map and in Bode diagrams. Consequently, these graphical representations can be applied to the demodulation of the interference pattern, making it possible to perform measurements in the complex s-plane.

## 4. Discussion

This paper corroborates that the cavity length variation within the cavity of an interferometer can be graphed in the complex plane s by employing pole-zero maps, Bode plots, and a low-fineness Fabry–Pérot interferometer, thus, verifying the theory presented in Section 2. For the analysis, the modulated function $f_{m}\left(\lambda ,\theta \left(\Delta L\right)\right)$ is filtered from the interference pattern, eliminating the envelope function $f_{e}\left(\lambda \right)$. Afterward, the function $f_{m}\left(\lambda ,\theta \left(\Delta L\right)\right)$ is transformed into the complex function $F_{m}\left(s,\theta \left(\Delta L\right)\right)$ employing the Laplace transform, from which the pole-zero maps and Bode plots are elaborated; several cavity length values were deployed for the construction of these graphs. These graphic representations are in agreement with the theoretical analysis. Based on these results, the following points are worth mentioning:A disturbed Fabry–Pérot interferometer is analyzed in the complex plane s by graphically employing the pole-zero map and the Bode diagram;The pole-zero map can be deployed to detect cavity length variations. The data can be obtained from the position of the zeros (see Figure 4c, Figure 5c and Figure 6c). Table 2 displays the values of ΔL, $\theta \left(\Delta L\right)$, and the respective positions of the zeros;

Observing Table 2, varying the cavity length ΔL, the phase $\theta \left(\Delta L\right)$ is modified as well; consequently, the zeros undergo changes in their positions.
3.The pole-zero maps are employed for the measurement of disturbances in the interferometer cavity. Based on reference [5], when $\theta \left(\Delta L\right)=0$, a reference vector is defined from the origin until the position of the zero $s_{0}=27.565i$. Meanwhile, when $\theta \left(\Delta L\right)\neq 0$, a vector is defined by the perturbation from the origin to the position of the zero. Taking this into consideration along with Table 2 and reference [5], Table 3 is generated, where a detection vector is defined as follows;



(39)
$$s_{0}^{det}=s_{0}^{ref}-s_{0}^{per},$$


$s_{0}^{det}$ is the detection vector, $s_{0}^{ref}$ is the reference vector, and $s_{0}^{per}$ is the perturbation vector.

**Table 3 sensors-25-02182-t003:** Detection vector calculated due to cavity length changes.

$\Delta L\left[nm\right]$	$\mathrm{Reference},\;s_{0}^{ref}$	$\mathrm{Zero}\;\mathrm{Position},\;s_{0}^{per}$	$s_{0}^{det}=s_{0}^{ref}-s_{0}^{per}$
0	$s_{0}=27.565i$	$s_{0}=27.565i$	$s_{0}^{det}=0$
102.08	$s_{0}=27.565i$	$s_{0}=-9.24+29.1i$	$s_{0}^{det}=-9.24+1.535i$
204.26	$s_{0}=27.565i$	$s_{0}=-23.8+36.4i$	$s_{0}^{det}=-23.8+8.835i$

Note: $s_{0}^{ref}$ is the reference vector, $s_{0}^{per}$ is the perturbation vector, and $s_{0}^{det}$ is the detection vector.

4.Bode plots are deployed for the visualization of variation in the optic-path difference in interferometers;5.A Bode plot can be employed for cavity variation measurements;6.Changes in ΔL cause changes in phase $\theta \left(\Delta L\right)$; subsequently, the magnitude $\left|F_{m}\left(\omega ,\theta \left(\Delta L\right)\right)\right|$ changes as well; hence, phase $\emptyset \left(\omega ,\theta \left(\Delta L\right)\right)$ is affected accordingly (see Equations (27) and (28) along with Figure 4d, Figure 5d and Figure 6d);7.The Bode plots can be deployed for the demodulation of the interference patterns generated by the interferometers;8.The pole-zero maps and the Bode plots can be employed for the demodulation of signals in interferometric systems and quasi-distributed sensors where the local sensor is an interferometer.

On comparing our work with reference [5], this paper proposes a graphical study of cavity length variations in the complex plane s through pole-zero maps and Bode plots, whereas the referenced study proposes an analytical study. Our future line-of-work is aimed at procuring the following directions: (1) implementing an algorithm for demodulating the interference pattern based on pole-zero maps and Bode plots; (2) developing algorithms of signal demodulation in the complex plane s for quasi-distributed sensors; (3) deploying the pole-zero maps and Bode plots for the analysis of quasi-distributed sensors; (4) performing a noise analysis; and (5) eliminating ambiguity 2πM in the complex plane s.

## 5. Conclusions

This paper studies the cavity length variations, ΔL, caused by the disturbance above the cavity of an interferometer in the complex plane s employing pole-zero maps and Bode plots. This study was initially performed in a theoretical manner and was then corroborated through numerical experiments. Both theory and experimentation are in agreement. Based on the graphic results of the pole-zero map, the cavity variation, ΔL, can be observed in the position of the zeros while the position of the poles remains constant. Furthermore, observing the Bode plots, the cavity variation, ΔL, modifies the magnitude and phase of the complex function. Both graphic methods can be deployed in the demodulation of interferometer signals and quasi-distributed sensors.

For this analysis, the data of the envelope function $f_{e}\left(\lambda \right)$ is not considered, given that it does not contain relevant information regarding the disturbance on the interferometer. Consequently, for this study, only the modulated function $f_{m}\left(\lambda ,\theta \left(\Delta L\right)\right)$ was employed since it contains information regarding the cavity length variations, ΔL. This simplified the theoretical analysis and the graphic representation of the function $F_{m}\left(s,\theta \left(\Delta L\right)\right)$ through pole-zero maps and Bode plots.

## Figures and Tables

**Figure 1 sensors-25-02182-f001:**
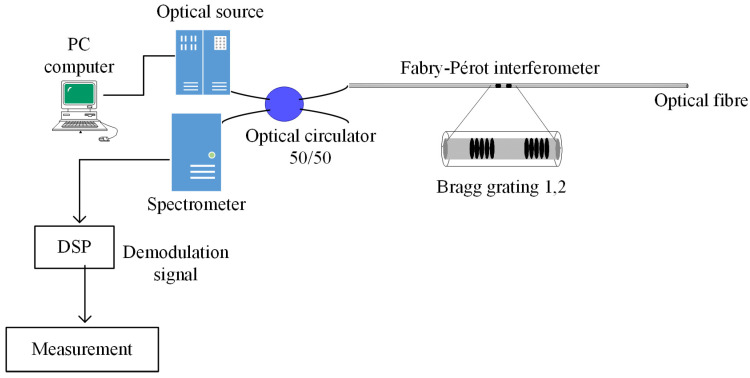
Optical system employed for the low-fineness Fabry–Pérot interferometer.

**Figure 2 sensors-25-02182-f002:**
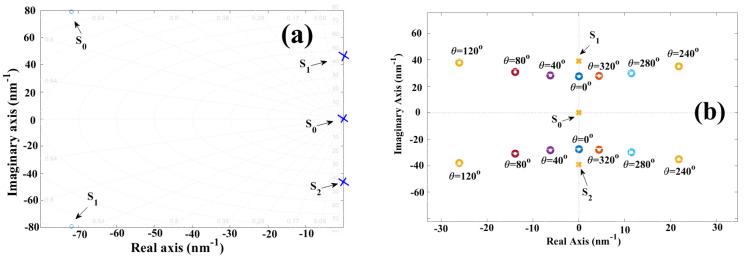
Pole-zero diagrams: (**a**) when the Fabry-Pérot interferometer is perturbed externally and the phase is $\theta \left(\Delta L\right)=50^{{}^{\circ}}$; (**b**) when the interferometer is externally disrupted by a variable causing the following phases: $\theta \left(\Delta L\right)=0{}^{\circ}$, 40°, 80°, 120°, 240°, 280°, and 320°.

**Figure 3 sensors-25-02182-f003:**
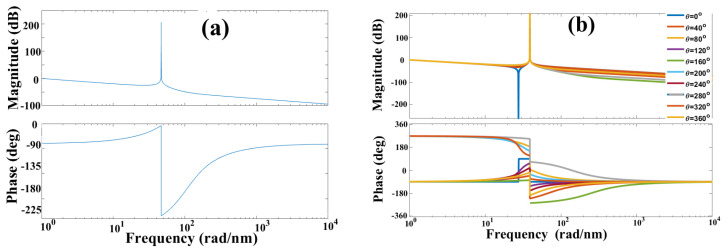
Bode diagram of the interference pattern: (**a**) when the interferometer cavity is perturbed $\theta \left(\Delta L\right)=50{}^{\circ}$, and (**b**) when the interferometer is perturbed by an external agent and the phase $\theta \left(\Delta L\right)$ value is different: 0°, 40°, 80°, 120°, 160°, 200°, 240°,  280°, 320°, and 360°.

**Figure 5 sensors-25-02182-f005:**
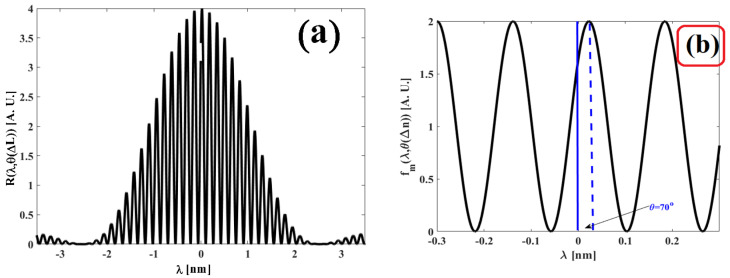
Obtained graphs for the refractive index variation ΔL=157.0129 μm, $\left(\theta (\Delta L)=70^{{}^{\circ}}\right)$: (**a**) Interference pattern ${\left.R\left(\lambda ,\theta \left(\Delta L\right)\right)\right\rfloor }_{\theta \left(\Delta n\right)=70^{{}^{\circ}}}$ produced by the Fabry–Pérot interferometer; (**b**) the function ${\left.f_{m}\left(\lambda ,\theta \left(\Delta L\right)\right)\right|}_{\theta \left(\Delta L\right)=70^{{}^{\circ}}}$ is filtered from the interference pattern; (**c**) pole-zero map obtained for ${\left.F_{m}\left(s,\theta \left(\Delta L\right)\right)\right|}_{\theta \left(\Delta L\right)=70^{{}^{\circ}}}$; and (**d**) the Bode diagram obtained for ${\left.F_{m}\left(s,\theta \left(\Delta L\right)\right)\right|}_{\theta \left(\Delta L\right)=70^{{}^{\circ}}}$.

**Figure 6 sensors-25-02182-f006:**
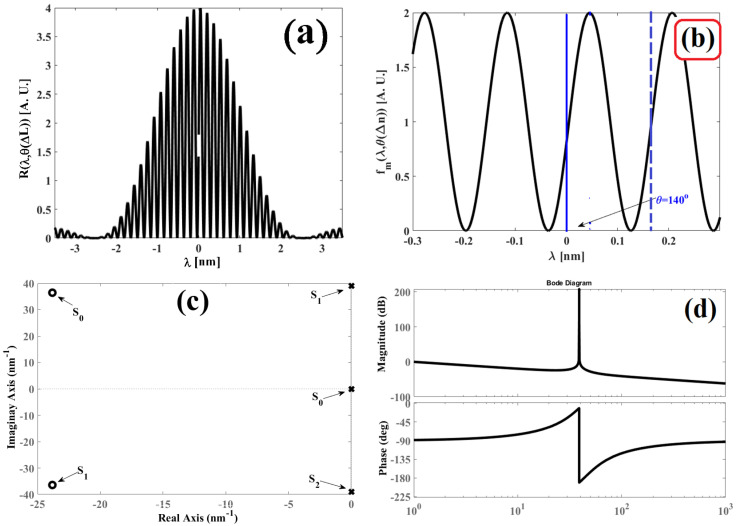
Obtained graphs for the cavity length variation ΔL=314.205 μm, $\left(\theta (\Delta L)=140^{{}^{\circ}}\right)$: (**a**) Interference pattern ${\left.R\left(\lambda ,\theta \left(\Delta L\right)\right)\right\rfloor }_{\theta \left(\Delta L\right)=140^{{}^{\circ}}}$ produced by the Fabry–Pérot interferometer; (**b**) function ${\left.f_{m}\left(\lambda ,\theta \left(\Delta L\right)\right)\right\rfloor }_{\theta \left(\Delta L\right)=140^{{}^{\circ}}}$ filtered from the interference pattern; (**c**) pole-zero map obtained for function ${\left.F_{m}\left(s,\theta \left(\Delta L\right)\right)\right\rfloor }_{\theta \left(\Delta L\right)=140^{{}^{\circ}}}$; and (**d**) the Bode diagram obtained for function ${\left.F_{m}\left(s,\theta \left(\Delta L\right)\right)\right\rfloor }_{\theta \left(\Delta L\right)=140^{{}^{\circ}}}$.

**Table 1 sensors-25-02182-t001:** Comparison of experimental results shown in Figure 4b–d, Figure 5b–d and Figure 6b–d.

Cavity Length Variation ΔL (nm)	$\mathrm{Phase}\;\theta \left(\Delta L\right)$ (°)	Poles and Zeros Map		Bode Diagrams	
		Poles (nm^−1^)	Zeros (nm^−1^)	Asymptotic	Phase Changes
0	$0^{{}^{\circ}}$	$\begin{array}{c} s_{0}=0\\ s_{1}=38.983i\\ s_{2}=-38.983i \end{array}$	$\begin{array}{c} s_{0}=27.565i\\ s_{1}=-27.565i \end{array}$	2	2
102.08	$70^{o}$	$\begin{array}{c} s_{0}=0\\ s_{1}=38.983i\\ s_{2}=-38.983i \end{array}$	$\begin{array}{c} s_{0}=-9.24+29.1i\\ s_{1}=-9.24-29.1i \end{array}$	1	1
204.26	$140^{o}$	$\begin{array}{c} s_{0}=0\\ s_{1}=38.983i\\ s_{2}=-38.983i \end{array}$	$\begin{array}{c} s_{0}=-23.8+36.4i\\ s_{1}=-23.8-36.4i \end{array}$	1	1

**Table 2 sensors-25-02182-t002:** Position changes due to differences in cavity length.

$\Delta L\left[nm\right]$	$\theta \left(\Delta L\right)$	Zero Positions	Experiment
0	$0^{{}^{\circ}}$	$\begin{array}{c} s_{0}=27.565i\\ s_{1}=-27.565i \end{array}$	1 (Section 3.1)
102.08	$70^{{}^{\circ}}$	$\begin{array}{c} s_{0}=-9.24+29.1i\\ s_{1}=-9.24-29.1i \end{array}$	2 (Section 3.2)
204.26	$140^{{}^{\circ}}$	$\begin{array}{c} s_{0}=-23.8+36.4i\\ s_{1}=-23.8-36.4i \end{array}$	3 (Section 3.3)

## Data Availability

The data related to the results that support our conclusions are available upon request from the authors. This can be carried out via e-mail. We will be pleased to respond.

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
