# Peer review of "Graphical Representation of Cavity Length Variations, ΔL, on s-Plane for Low-Finesse Fabry–Pérot Interferometer"

_sensors, 2025, doi:10.3390/s25072182_

Round 1

Reviewer 1 Report

Comments and Suggestions for Authors

This manuscript presents a study of cavity length variations that could be impacted by disturbances in the vicinity of cavities of an interferometer, using theoretical (Bode) and numerical investigations. The manuscript is well organized and the studies seem sound, scientifically and technically. The results (Bode plots and numerical analysis) is plausible based on the problem setting and parameters employed. For these reasons, I recommend publication, assuming the authors will address comments by the rest of the reviewers.

Author Response

Guadalajara, Jalisco, México

March 21, 2025

Dear Editor in Chief:

Sensors

I am pleased to resubmit for publication the revised version of our manuscript entitled: “Graphical representation of cavity length variations  on s-plane for a low-finesse Fabry Perot Interferometer (Manuscript ID: sensors-3511829) for publication. I would like to express my gratitude to the Editor and reviewers for their thorough and constructive feedback. In response to their valuable suggestions and comments, we have carefully revised the manuscript, and we believe these revisions have significantly improved the quality of the paper.

In the revised manuscript, the reviewers' comments have been marked in bold, changes to the text are highlighted in blue, and deleted text is marked in red.

Review´s comments:

Review 1

Comments and Suggestions for Authors

This manuscript presents a study of cavity length variations that could be impacted by disturbances in the vicinity of cavities of an interferometer, using theoretical (Bode) and numerical investigations. The manuscript is well organized and the studies seem sound, scientifically and technically. The results (Bode plots and numerical analysis) is plausible based on the problem setting and parameters employed. For these reasons, I recommend publication, assuming the authors will address comments by the rest of the reviewers.

Answer:

We sincerely appreciate the positive feedback and recommendation for publication.

Thank you very much for your kind attention. We hope you find our manuscript suitable for publication and look forward to hearing from you soon.

Sincerely:

Dr. José Trinidad Guillen Bonilla

Departamento de Electro-fotónica, CUCEI.

Universidad de Guadalajara,

Blvd- M. García Barragan 1421, Guadalajara, Jalisco,

  1. P. 44410, México.

e-mail: trinidad.guillen@academicos.udg.mx

Tel.: +52 (33) 1378 5900 (ext.

Reviewer 2 Report

Comments and Suggestions for Authors

The manuscript itself could be interesting for the readers, but it lacks a few important points. In my opinion, the manuscript should undergo the major revision prior to the publication. Below are a few issues both minor and major I've detected.

Major issues

The reference list is rather sparse. In my opinion, it has to be improved.

Fig. 2 is not informative and it does not contain the axis labels - what is the values range?
The problem of the absence of the values range is the same for the Fig. 3

Fig. 3 should be redesigned and explained in details. Now it is unclear what the authors are supposed to present. Probably, the color scheme should be redesigned also, since now the "blue" curve for phase seems wrong (it goes back and forth)

Fig. 3a and Fig. 4c duplicates each other. I would recommend either to reorganize these parts of the manuscript in order not to duplicate the figures or to make appropriate references to the previously provided figures. The first case is preferrable. Especially taking into account, that after the fig. 4 the authors are talking about the comparison of fig. 2a, 3a, 4c, 4d. It is better to organize the manuscript in such a way that the reader is not scrolling back and forth to compare the images.

What is the meaning of the cavity disturbance values for experiment 2 and 3 (157,012 and 314,205 um, 70 and 140 degrees)? Why do the authors chose these particular values and to what real circumstances they correspond to?

The analysis and comparison section of the results for the experiments 1-3 are missing, in my opinion. More analysis is needed, including the comparison of the resultant figures graphs with interference patterns, pole-zero maps as well as Bode plots.

Minor issues
- Page 2 - "lineal transformation" - I believe it should be "linear". If yes, please, correct
- Page 2 - "to study the behaviour of the system under study" - please, rephrase the sentence and avoid words duplication
- Page 2 - last sentence - should be "based on", please, correct
- Name of section 2 - what is "s" here, is it a symbol of a plane or a typo? If it is a symbol of a plane, please format it appropriately so that it does not look like a plane text. The same is for the "s" symbols in the text of the manuscript. Please, pay attention to the formatting of the special symbols.
- Equation (1) - the description starts with the wrong parameter - R(lambda, delta n) instead of R(lambda, delta L) - please, correct
- There are a few other typos, both grammatical and punctuational (extra spaces, etc.)

Comments on the Quality of English Language

The English language is in general good, but the manuscript should be carefully checked again, since there are a few typos and mistakes

Reviewer 3 Report

Comments and Suggestions for Authors

The equation 1 and 2 are wrong at expressing the term of Cos-function and (ΔL), because of dimensional inhomogeneity. This mistake often emerges also in the text or other equations that need to be examined completely.

The fineness value of the so-called “low-fineness Fabry–Pérot interferometer” should be evaluated and expressed quantitatively, not only the value of reflectivity described.

Round 2

Reviewer 2 Report

Comments and Suggestions for Authors

The authors have addressed all the issues. In my opinion, the manuscript is ready for the publication